# Genetic Foundation of Male Spur Length and Its Correlation with Female Egg Production in Chickens

**DOI:** 10.3390/ani14121780

**Published:** 2024-06-13

**Authors:** Anqi Chen, Xiaoyu Zhao, Xiurong Zhao, Gang Wang, Xinye Zhang, Xufang Ren, Yalan Zhang, Xue Cheng, Xiaofan Yu, Huie Wang, Menghan Guo, Xiaoyu Jiang, Xiaohan Mei, Guozhen Wei, Xue Wang, Runshen Jiang, Xing Guo, Zhonghua Ning, Lujiang Qu

**Affiliations:** 1National Engineering Laboratory for Animal Breeding, College of Animal Science and Technology, China Agricultural University, Beijing 100193, China; 18800017599@163.com (A.C.); zxiurong_feign@163.com (X.Z.); wanggang@cau.edu.cn (G.W.); xinye_leaf@163.com (X.Z.); rxf1828@163.com (X.R.); 15901003721@163.com (Y.Z.); chengxue@cau.edu.cn (X.C.); b20233040354@cau.edu.cn (X.Y.); sy20233040865@cau.edu.cn (M.G.); jxy1581908148@163.com (X.J.); 2022333020320@cau.edu.cn (X.M.); ningzhh@cau.edu.cn (Z.N.); 2Xingrui Agricultural Stock Breeding, Baoding 072550, China; xrnmyfzx@bdxrkj.com; 3Xinjiang Production and Construction Corps, Key Laboratory of Protection and Utilization of Biological Resources in Tarim Basin, Tarim University, Alar 843300, China; whedky@126.com; 4Qingliu Animal Husbandry, Veterinary and Aquatic Products Center, Sanming 365501, China; 17866705322@163.com; 5VVBK Animal Medical Diagnostic Technology (Beijing) Co., Ltd., Beijing 100199, China; vvetwangxue6307@sina.com; 6College of Animal Science and Technology, Anhui Agricultural University, Hefei 230036, China; jiangrunshen@ahau.edu.cn (R.J.); guoxing0405@126.com (X.G.)

**Keywords:** chicken, spur, genetics, heritability, Pool-GWAS

## Abstract

**Simple Summary:**

The spur refers to the protrusion near the tarsometatarsus on both sides of the calves of chickens, and their length varies from individual to individual. By analyzing phenotypic measurement results, we found that there were significant variations in spur length in layers. After calculating heritability, we found that spur length had a high level of heritability in the population, which also proved that this quantitative trait was controlled by multiple genes. Subsequently, we utilized genomic sequencing to identify candidate genes associated with spur length. As is known to all, the genetic material passed from parents to all siblings is similar, but some gender-biased traits such as spurs and egg-laying only show in one gender. We also attempted to find a connection between these two traits in this article.

**Abstract:**

Spurs, which mainly appear in roosters, are protrusions near the tarsometatarsus on both sides of the calves of chickens, and are connected to the tarsometatarsus by a bony core. As a male-biased morphological characteristic, the diameter and length of spurs vary significantly between different individuals, mainly related to genetics and age. As a specific behavior of hens, egg-laying also varies greatly between individuals in terms of traits such as age at first egg (**AFE**), egg weight (**EW**), and so on. At present, there are few studies on chicken spurs. In this study, we investigated the inheritance pattern of the spur trait in roosters with different phenotypes and the correlations between spur length, body weight at 18 weeks of age (**BW18**), shank length at 18 weeks of age (**SL18**), and the egg-laying trait in hens (both hens and roosters were from the same population and were grouped according to their family). These traits related to egg production included **AFE**, body weight at first egg (**BWA**), and first egg weight (**FEW**). We estimated genetic parameters based on pedigree and phenotype data, and used variance analysis to calculate broad-sense heritability for correcting the parameter estimation results. The results showed that the heritability of male left and right spurs ranged from 0.6 to 0.7. There were significant positive correlations between left and right spur length, BW18, SL18, and BWA, as well as between left and right spur length and AFE. We selected 35 males with the longest spurs and 35 males with the shortest spurs in the population, and pooled them into two sets to obtain the pooled genome sequencing data. After genome-wide association and genome divergency analysis by F_ST_, allele frequency differences (**AFDs**), and XPEHH methods, we identified 7 overlapping genes (CENPE, FAT1, FAM149A, MANBA, NFKB1, SORBS2, UBE2D3) and 14 peak genes (SAMD12, TSPAN5, ENSGALG00000050071, ENSGALG00000053133, ENSGALG00000050348, CNTN5, TRPC6, ENSGALG00000047655,TMSB4X, LIX1, CKB, NEBL, PRTFDC1, MLLT10) related to left and right spur length through genome-wide selection signature analysis and a genome-wide association approach. Our results identified candidate genes associated with chicken spurs, which helps to understand the genetic mechanism of this trait and carry out subsequent research around it.

## 1. Introduction

Sexual dimorphism refers to the differences between sexes of the same species [1,2], particularly in terms of color [3], body size [2,4,5], song [6], and even embryo [7] between males and females. As a sexually dimorphic trait in chickens, spurs are generally present in males after sex maturation and can be found in some adult hens [8,9,10], but are not common. Spurs are located below the calves on both sides and above the feet, connected to the tarsometatarsus by a bony core, and can grow with age [8]. Spur length is believed to be related to lighting stress [11], hormone levels [12,13,14], and growth conditions [15], and may play a certain role in individual viability [16,17], mate choice [16,18], and reproduction [16,17]. In the past, many people thought of the spur as the toe of a chicken, but its diameter was much larger than the toes, which could be considered as the protruding part of the shank. The lifestyle of birds has undergone certain changes during evolution, such as changes in feather structure and color, to adapt to different living environments [19]. The existence of spurs contributes to individual survival, so we believe that spurs will evolve in length and shape to adapt to different living environments.

Spurs, along with some lower leg bones, is often regarded as a cultural relic with a long history in China. The length, color, and shape of the spur are important factors that affect its own value. In some Chinese local markets, spur length is related to its economic value, since some people consider longer spurs a symbol of good health and quality of the chicken. As an important trait often overlooked by breeders, spurs have not been systematically selected, so spur length has significant phenotypic variation in most chicken breeds. In our study, we selected the Rhode Island Red (RIR) chicken to measure the left and right spur length of 406 adult male chickens, and estimated the genetic parameters of the trait based on pedigree data and variance analysis. Subsequently, we conducted a correlation analysis on left and right spur length, body weight at 18 weeks of age (BW18), shank length at 18 weeks of age (SL18) of roosters, as well as related hens’ egg production traits, including the age at first egg (AFE), body weight at first egg (BWA), and first egg weight (FEW). Finally, we performed a genome-wide association study (Pool-GWAS) and genome divergency analysis to identify candidate genes that could affect spur length. The above results supplemented the research on spurs in chickens and provided the possibility for systematic breeding for spur length.

## 2. Materials and Methods

### 2.1. Ethics Statement

This study was conducted following the guidelines for experimental animals established by the Animal Care and Use Committee of China Agricultural University.

### 2.2. Animal and Chicken Phenotypic Measurement

A total of 406 roosters of RIR chickens were selected in this study. All birds were given the same feed conditions in single cages. At 50 weeks of age, we used a vernier caliper to measure left and right spur length which was along the top edge of the spur from the tarsometatarsus to its distal end [10,15] (Figure 1). In most cases, the spur trait was unique to roosters, while the egg-laying trait was specific to hens. We also recorded the growth traits (BW18, SL18) of these roosters and egg production data (AFE, BWA, and FEW) of hens from the same generation (same strain and batch) and raised in the same conditions as these roosters. The calculation of correlation between traits was often carried out within the same individual. The same trait had different genetic bases within different families, so phenotypic differences between individuals were mainly caused by the bases. Compared with phenotypic variations caused by inter-family differences, we believed that the phenotypic variation caused by genetic variations within the same family could be ignored. Therefore, within the same family, a similar phenotype could generally be exhibited regardless of gender, although certain traits only appear in specific genders. For the calculation of the correlation between gender-specific traits in the same population, we planned to merge the male and female data within the same family for analysis. To calculate the correlation between left and right spur length and egg production traits, we divided the hens and roosters into one group based on their families, that is, all half siblings were grouped into one family. Then, we used SPSS 26 software to calculate the correlation between these traits. Finally, we calculated the average value of left and right spur length for each chicken, and used this average as the sole criterion (spur length) for individual comparison and sample selection.

### 2.3. Genetic Parameter Estimation

Genetic parameters based on pedigree for left and right spur length were estimated using a Multi-trait Animal Model in DMUv6 software [20] as in the following formula.
y = Xμ + Za + e

In the formula, y is the phenotypic value, X is the matrix of fixed effects, μ is a vector composed of fixed effects, Z is the correlation matrix of random effects, a is a vector composed of random effects, and e represents the random residual effects. Using software, we could estimate the variance and covariance components with the average information restricted maximum likelihood method (AI-REML). In our study, all roosters were from the same population and raised in the same conditions, so no fixed effects were considered in this model. However, ignoring the potential impact of environment or other factors on traits could to some extent affect the accuracy of estimated results.

We also used variance analysis to calculate genetic variance and residuals in R 4.3.1 software, and obtained broad-sense heritability based on the following formula, which served as a reference for the results based on pedigree estimation.
h^2^ = V_G_/(V_G_ + V_E_)
V_E_ = ME_e_
V_G_ = MS_G_ − MS_e_/r

In the formula, h^2^ is broad-sense heritability, V_G_ is genetic variance, V_E_ is residual, ME_e_ is mean-square error, and MS_G_ is mean-square genetic variance. V_G_ and V_E_ can be unbiasedly estimated by MS_G_ and MS_E_.

### 2.4. Genomic Analysis

As mentioned in Section 2.2, we used the average value of left and right spur length (spur length) as the sole indicator for individual selection, and then selected 35 individuals with the longest spur length and 35 with the shortest spur length for blood collection and sequencing analysis. Among the 35 hens with the longest spur length, the average value was 25.13 mm, the maximum value was 31.62 mm, and the minimum value was 23.70 mm. Among the 35 hens with the shortest spur length, the average value was 15.37 mm, the maximum value was 16.52 mm, and the minimum value was 11.42 mm. Blood samples were collected from the brachial vein and DNA was extracted using a FlaPure Animal Tissue/Cell/Blood DNA Extraction Kit (Genesand Biotech Co., Ltd., Beijing, China). Two pools for pool-sequencing were produced by Illumina Novaseq 6000 with 150 bp paired-end (Biomarker Technologies Corporation, Beijing, China). The raw data were processed in the first step through including a cutting adapter, with removing low-quality bases. Then, we compared the data to the reference genome (Gallus gallus 6.0) using a Burrows–Wheeler Aligner (BWA, version 0.7.17) [21]. Samtools (v.1.16.1) [22] and Picard tools (v2.25.2) (https://broadinstitute.github.io/picard/, accessed on 15 May 2024) were used to convert and clear duplicates. The “mpileup” option was executed to merge BAM files, and then the “mpileup2sync.pl” file was used to synchronize the mpileup file with PoPoolation2 [23]. In addition, we also used a Genome Analysis Toolkit (GATK, version 3.8) [24] for re-alignment, Base Quality Score Recalibration, and SNP calling. After performing hard filtration in GATK with the following conditions: QUAL < 30.0 || QD < 2.0 || FS > 60.0 || MQ < 40.0 || MQRankSum < −12.5 || ReadPosRankSum < −8.0, we obtained a total of 5,775,816 SNPs.

Pool-GWAS, FST, and AFD were calculated by PoPoolation2 (https://sourceforge.net/projects/popoolation2/, accessed on 15 May 2024) by three files using existing files in the software. The Cochran–Mantel–Haenszel (CMH) test is a hierarchical chi-square test used to detect an association between SNPs and traits [25]. To correct the testing errors, we performed an FDR control for CMH test using the fdrtool package (https://cran.r-project.org/web/packages/fdrtool/index.html, accessed on 15 May 2024) in R 4.3.1 software. We set the SNPs with FDR values less than 1 × 10^−16^ as candidate SNPs. XPEHH was calculated by selscan [26]. In addition, the windows or SNPs within the top 0.1% values for the FST and AFD and 1% for XPEHH were considered as the candidate regions and SNPs. The SNPs and regions obtained from above method were annotated in BioMart of Ensembl to obtain gene stable ID, gene name, and candidate gene (http://www.ensembl.org/biomart/martview, accessed on 15 May 2024). Then, we used Gene Ontology (GO) enrichment analysis and Kyoto Encyclopedia of Genes and Genomes (KEGG) analysis in KOBAS to gain gene function (http://bioinfo.org/kobas/, accessed on 15 May 2024) [27].

## 3. Results

### 3.1. Heritability and Correlation Analysis

We demonstrated a method for measuring spur length (Figure 1), as detailed in Section 2. The average values of left and right spur length within the rooster population were 19.48 ± 3.09 mm and 20.21 ± 3.10 mm, respectively. In terms of measuring length, the left and right spur length were close.

We calculated broad-sense heritability of right and left spur length using the formula given in Section 2.3, which was 0.72 and 0.67, as shown in Table 1. Heritability based on the pedigree of right and left spur length was 0.87 and 0.77, with an SE value of 0.15. The results obtained by adjusting heritability based on pedigree with SE were similar to those obtained by variance analysis, so we believed that it was more appropriate to consider the heritability of left and right spur length as 0.6–0.7. The above results indicated that the development degree of the left and right spur length was similar and they were all high-level heritable traits. We also analyzed the correlation between left and right spur length in roosters and AFE, BWA, and FEW in hens, as shown in Table 1, as detailed in Section 2. The correlation between right and left spur length was significant and positive (r2 = 0.844 **), and both were significantly positively correlated with BW18 (r2 = 0.224 ** for the right spur length, r2 = 0.173 for the left), SL18 (r2 = 0.178 ** for the right spur length, r2 = 0.190 ** for the left), AFE (r2 = 0.101 ** for the right spur length, r2 = 0.100 ** for the left), and BWA (r2 = 0.069 * for the right spur length, r2 = 0.069 * for the left). The correlation between BW18 and SL18 was 0.495 **, the correlation between BW18 and BWA was 0.191 **, and the correlation between SL18 and BWA was 0.107 **.

### 3.2. Pool-GWAS and Genome-Wide Selection Signature Analysis

After quality filtering steps, we used the CMH test to detect the SNPs for allele frequency difference between long spur and short spur chickens. We found a total of 6620 SNPs that were significantly associated with spurs (at a False Discovery Rate (FDR) cutoff of 1 × 10^−16^, *p* value < 5.42 × 10^−20^) and annotated 728 genes (Appendix A). We also used FST, AFD, and XPEHH between the two pools of data. Finally, we separately found 104 regions, 10,492 SNPs, and 7369 SNPs that were significantly associated with spurs by the three methods, and identified 32, 944, and 351 genes (Appendix A). After enrichment analysis of all genes obtained from the four methods, we found that some genes could be classified based on GO terms. For example, 267 genes played a role in cytosol, and 221 genes played a role in the plasma membrane (Appendix A). Some genes were grouped based on KEGG pathways, including 37 and 42 involved in Focal adhesion and MAPK signaling pathways (Appendix A).

A total of seven overlapping genes selected through the four methods were centromere protein E (CENPE), FAT atypical cadherin 1 (FAT1), family with sequence similarity 149 member A (FAM149A), mannosidase beta (MANBA), nuclear factor kappa B subunit 1 (NFKB1), sorbin and SH3 domain-containing 2 (SORBS2), and ubiquitin conjugating enzyme E2 D3 (UBE2D3) (Figure 2). The consequence of variation at these loci associated with spurs in five genes included the intron variant, splice acceptor variant, synonymous variant, missense variant, and so on (Appendix A). In a previous study, it was found that CENPE might play a role in centromere [28,29] and kinetochore [30,31] mechanisms, FAT1 might play a role in the maintenance of retinal cells [32] and the stimulation and differentiation of primary chicken myoblasts [33], and UBE2D3 was a candidate gene in music abilities [34]. The expression of the NFKB1 gene was associated with the addition of quercetin to alleviate chicken duodenal inflammation [35], chicken infection with Eimeria coccidiosis [36], the addition of organic acid and botanical vanillin to reduce chicken clinical necrotic enteritis (NE) [37], and the addition of organic acid-based formulation (OABF) to improve chicken gut ecology [38]. Epidermal growth factor (EGF) stimulated the proliferation of chicken primordial germ cells (PGCs) via activation of Ca^2+^/PKC involving the NFKB1 signaling pathway [39].

We also annotated several peak SNPs in Pool-GWAS, FST, and XPEHH. There were three SNPs on chromosome 2, three SNPs on chromosome 4, and three SNPs on chromosome 11 by Pool-GWAS that reached a certain level of significance (*p* value < 1 × 10^−46^), six regions on chromosomes 1 and 8 had a peak in FST (FST > 0.20), six SNPs on chromosome 2 and three SNPs on chromosome 5, and Z had a peak in XPEHH (XPEHH > 4.2) (Appendix A). We found 14 peak genes, including sterile alpha motif domain-containing 12 (SAMD12), tetraspanin 5 (TSPAN5), ENSGALG00000050071, ENSGALG00000053133, ENSGALG00000050348, contactin 5 (CNTN5), transient receptor potential cation channel subfamily C member 6 (TRPC6), ENSGALG00000047655, thymosin beta 4X (TMSB4X), limb and CNS expressed 1 (LIX1), creatine kinase B (CKB), nebulette (NEBL), phosphoribosyl transferase domain-containing 1 (PRTFDC1), and myeloid/lymphoid or mixed-lineage leukemia, translocated to 10 (MLLT10) (Figure 2). In a previous study, it was found that CKB might play a role in the response of chickens infected with Salmonella [40,41] and feed efficiency in chickens [42]. LIX1 was locally expressed during early chicken limb development [43], and the expression of TMSB4X underwent certain changes during maturation of hair cells [44]. TRPC6 might play a role in regulating calcium influx [45,46].

## 4. Discussion

### 4.1. Genetic Parameter Estimates

Quigley et al. (1951) compared spur length in the cockerel, slip, and capon at 9, 11, and 14 weeks of age and found that spur length was capon > slip > cockerel in all records, indicating that spur length responded rapidly to a reduction in testicular secretions and a possible quantitative relation between the growth capacity of the spur at the ages investigated and the restraining action of the gonad [47]. In our study, we estimated the genetic parameters of spur traits for roosters in RIR chickens, and found that the heritability levels of left and right spur length were fairly high, both between 0.6 and 0.7. As mentioned in Section 2.3, ignoring possible environmental or other factors could lead to overestimation of the results. Fairfull et al. (1986) found that there were no significant differences between left and right spur lengths, the incidence of spurs varied significantly in different strains, and the heritability of spur length was overall about 0.4 [10]. We also found significant positive correlations between left and right spur length, BW18, SL18, and BWA, and a significant positive correlation between left and right spur length and AFE. The above results might be used to explain similar development within the same family of roosters and hens. The study also reported a certain correlation between the left and right spur length of hens and traits such as egg production, egg weight, and Haugh units [10]. The relevant research results indicated that spur length was a highly heritable trait, and suggested a possibility of correlation between spur length and production traits, which greatly supported our findings.

### 4.2. Gene Function Analysis

In our study, we used two sets of pool-sequencing data from two groups of individuals with two phenotypes (long spur, short spur), and identified seven overlapping genes associated with spur length, including CENPE, FAT1, FAM149A, MANBA, NFKB1, SORBS2, and UBE2D3. Li et al. (2024) found that CENPE was the key gene linked to the hepatotoxicity induced by Zearalenone (ZEN) exposure in broiler chickens, and its expression level showed significant differences between the treatment and control group [48]. The protein products of the CENP gene family were the components that made up the centromere, so changes in one of the protein products might cause changes in the centromere [28,29,30,31]. A study found that abnormal assembly of the kinetochore protein occurring in CENP-A-depleted cells indicates mislocalization of inner kinetochore proteins [28]. CENP-T elongation might lead to changes in the shape of the inner kinetochore [30], and CENP-T depends on CENP-N for chromosome localization [31]. CENP-O and CENP-R might regulate the size of the inner kinetochore without influencing the assembly of the outer kinetochore [31]. Li et al. (2022) found that FAT1 was the candidate target gene of miR-29b-1-5p, and miR-29b-1-5p could inhibit the proliferation of primary chicken myoblasts and stimulate their differentiation [33]. The UBE2D3 gene was a key candidate gene for musical ability [34].

Some candidate genes located on the peak should not be ignored. We obtained the peak locus or regions from Pool-GWAS, FST, and XPEHH by setting new thresholds, and annotated 14 peak genes, including CKB, CNTN5, LIX1, MLLT10, NEBL, PRTFDC1, SAMD12, TMSB4X, TRPC6, TSPAN5, and several unnamed genes. Mariani et al. (2001) identified a genomic region that might be associated with resistance in chickens infected with Salmonella, which was close to the CKB gene [40]. After infection with Salmonella Enteritidis, the expression of CKB in chicken serum decreased [41]. In broilers infected with coccidiosis, the expression of the MLLT10 gene was upregulated [49]. CKB and CKA are two melatonin receptors [50]. The expression of CKB has been found to be related to feed efficiency (FE) [42]. LIX1 was transiently expressed in the nascent hindlimb bud between Hamburger–Hamilton stages 15 and 19, and the transcript of LIX1 was also found in multiple tissues such as the basal plate of rhombomeres 3 and 5, pharyngeal, foregut mesenchyme, and so on [43]. Nie et al. (2016) found that the LIX1 gene was associated with the color of chickens’ earlobe [51]. Chlorogenic acid (CGA) could regulate the intestinal microbiota, barrier function, and immune function in chickens, and the expression of TMSB4X increased after CGA treatment [52], while the expression of TMSB4X decreased during differentiation of hair cells [44]. Qiao et al. (2019) induced ascites syndrome (AS) in chickens by injecting cellulose particles, and found a significant increase in intracellular calcium levels and downregulation of TRPC1 gene expression in the TRPC family in cardiomyocytes from AS chickens, suggesting that an increase in intracellular free calcium ion concentration could inversely regulate the expression of calcium channels [45].

## 5. Conclusions

In our study, we found that both left and right spur length had a high level of heritability and there were significant positive correlations between left and right spur length, BW18, SL18, and BWA, as well as between left and right spur length and AFE. By genomic analysis, 21 genes were considered candidate genes for spur length. Our results filled the gap in research related to spurs in chickens, identifying some loci and genes that could affect the trait, which provides assistance for subsequent systematic breeding.

## Figures and Tables

**Figure 1 animals-14-01780-f001:**
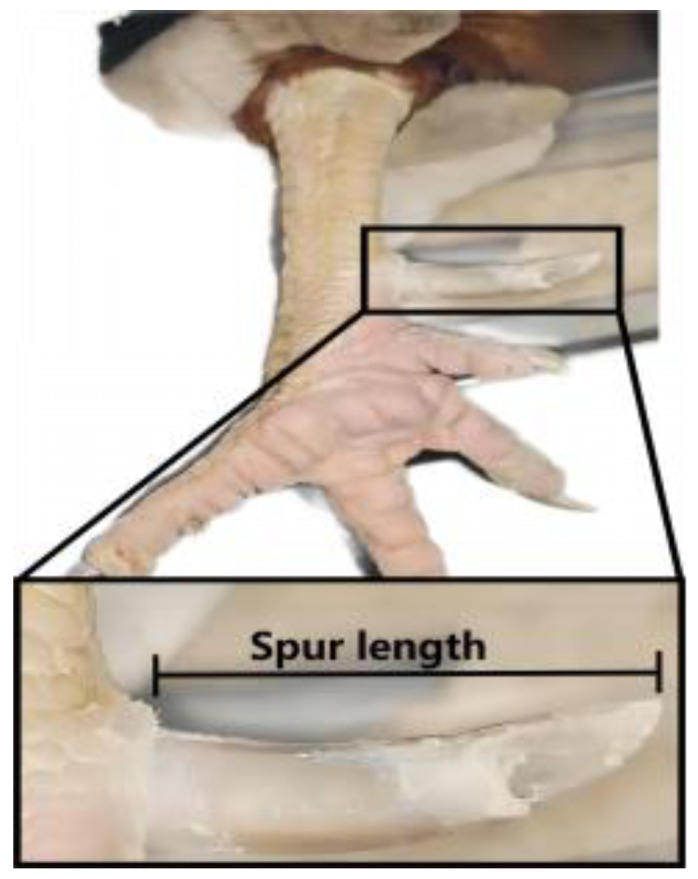
The location of the spur and method to measure spur length.

**Figure 2 animals-14-01780-f002:**
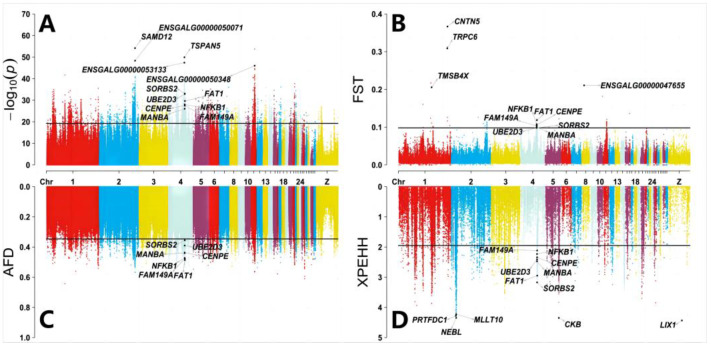
Manhattan plot and Q–Q plot for spur length by Pool-GWAS (**A**), FST (**B**), AFD (**C**), and XPEHH (**D**). Each point in the graph corresponds to the SNPs or regions in the dataset. The line in (**A**) represents the FDR value less than 1 × 10^−16^ (*p* value < 5.42 × 10^−20^), the lines in (**B**–**D**) represent the FST value of 0.098, the AFD of 0.346, and the XPEHH value of 1.94. The vertical axis (*y*-axis) of the Manhattan plot represents −log10 observed *p*-values of SNPs and the FST, AFD, and XPEHH values of SNPs compared between two groups (long spur, short spur). The horizontal axes (*x*-axes) all represent the position of these SNPs on the chromosome.

**Table 1 animals-14-01780-t001:** Heritability of male spur length and correlation between spur length, male growth traits, and female egg-laying traits.

Trait	Right Spur Length	Left Spur Length	BW18 ^1^	SL18 ^2^	AFE ^3^	BWA ^4^	FEW ^5^
Right spur length	0.72						
Left spur length	0.844 **	0.67					
BW18	0.224 **	0.173 **	-				
SL18	0.178 **	0.190 **	0.495 **	-			
AFE	0.101 **	0.100 **	−0.037	0.021	-		
BWA	0.069 *	0.069 *	0.191 **	0.107 **	0.241 **	-	
FEW	−0.008	0.013	−0.023	0.027	0.587 **	0.268 **	-

^1^ Body weight at 18 weeks of age. ^2^ Shank length at 18 weeks of age. ^3^ Age at first egg. ^4^ Body weight at first egg. ^5^ First egg weight. **, at the 0.01 level (double tailed), the correlation is significant; *, at the 0.05 level (double tailed), the correlation is significant. The values on the diagonal in this table are the heritabilities of each trait and the values in lower triangles are correlations.

## Data Availability

The pool-sequencing datasets are available with accession numbers SRR28731292 and SRR28731293.

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
