# Peer review of "Genetic Foundation of Male Spur Length and Its Correlation with Female Egg Production in Chickens"

_animals, 2024, doi:10.3390/ani14121780_

Round 1
Reviewer 1 Report
Comments and Suggestions for Authors The topic of your research was unexpected for me. I think that it is hardly advisable to use spur length as an indirect indicator to assess the productive qualities of a bird. Moreover, the correlation coefficients are low. You can determine live weight, egg weight and egg quality without any problems. However, if the length of the spur is really related to the resistance of chickens to a number of infectious and metabolic diseases, then the results of research in this direction will have good practical application.
Author Response
RE:Thank you for your comment. As is well known, the genetic material of all siblings is similar, but certain traits only manifest in specific genders. We grouped all individuals based on their families to determine the correlation between these traits in the population. This method provided a preliminary basis for determining the correlation between gender biased traits. And our results showed that there were significant positive correlations between the left and right spur length of roosters and the age at first egg and body weight at first egg of hens, as well as significant positive correlations between the shank length and body weight 18 week of age of roosters and body weight at first egg of hens. The above correlation results could be understood in other species, such as the offspring of taller parents (regardless of gender) generally being taller compared to the normal group, and the limb length of taller individuals was also longer compared to shorter individuals. We will further investigate the relationship between individual spur length and their resistance. In our research, we found a significant positive correlation between individual spur length and body weight at 18 weeks of age. Within a reasonable range, it was generally believed that individuals with significant physical weight tended to be in better health.
Reviewer 2 Report
Comments and Suggestions for Authors
Author Response
RE:Thank you for your comment. We mentioned in the abstract the methods of genome-wide selection signature analysis, please see line 48-49. We used different analysis methods to deeply explore overlapping and peak genes associated with spur length, and fully discussed the research results of these genes in existing articles. In the future, we will also focus on reliable minority genes for validation, and hope to combine our research with production.
Reviewer 3 Report
Comments and Suggestions for Authors
Please correct the references to the subchapters indicated in lines 131 and 175 (4.2 and 4.3).
Author Response
RE:Thank you for your comment. We made some modifications, please see line 93, 96, 119, 140, 178, 209, 267, 285.